# Low Cycle Fatigue of G20Mn5 Cast Steel Relation between Microstructure and Fatigue Life

**DOI:** 10.3390/ma15207072

**Published:** 2022-10-11

**Authors:** Antonin Bermond, Claire Roume, Jacques Stolarz, Matthieu Lenci, Jean-François Carton, Helmut Klocker

**Affiliations:** 1Mines Saint-Etienne, University Lyon, CNRS, UMR 5307 LGF, SMS Centre, F-42023 Saint Etienne, France; 2Safe Metal, 1 Boulevard de la Boissonnette, F-42110 Feurs, France

**Keywords:** G20Mn5 cast steel, columnar and equiaxed dendrites, micro-shrinkages, low-cycle fatigue, CT-scan, damage evolution

## Abstract

Cast steel is commonly used to produce structural and safety parts. Foundry processes allow producing parts from scrap steel directly to the required dimensions without any forming operation. Cast components may, however, exhibit macro- and micro-shrinkage porosities. The combined effect of macro- and micro-shrinkages on the fatigue behavior of cast steel has been characterized in the literature. Macro-shrinkages may nowadays be eliminated by adequate positioning of risers. However, micro-shrinkages will always be present in cast steel components. Present work addresses the influence of micro-shrinkage porosity on a G20Mn5 cast steel. G20Mn5 (normalized) ingots have been cast under industrial conditions, but ensuring the absence of macro-porosities. Solidification leads to two very different microstructures prior to the normalization treatment: columnar dendrites beneath the surface (**S**kin) and equiaxed microstructures close to the center (**C**ore). First, metallographic observations of the whole ingot revealed the same grain size in both areas. Fatigue samples were extracted, by differentiating two sampling volumes corresponding to columnar (S) and equiaxed solidification (C), respectively. The distribution of micro-porosities was determined on all samples by Micro-CT-scans. Core samples exhibit micro-porosities with volumes 1.7 larger than Skin samples. Low cycle fatigue tests (3 levels of fixed plastic strain) were run on both sample series (C, S). Results follow a Manson–Coffin law. **C**ore specimens exhibit lower fatigue life than Skin specimens. The differences in fatigue life have been related successfully to the differences in micro-porosities sizes.

## 1. Introduction

Cast steel is commonly used to produce structural and safety parts for trucks, railways and construction equipment. Steel foundry processes allow producing parts directly to the required dimensions without any forming operation. Moreover, it uses only scrap steel, so it is an ecological process that preserves Earth’s resources. A historical overview of casting may be found in [1]. The cast components of interest in the present work are submitted to fatigue loading.

Fatigue is damage resulting from cyclic loading. The fatigue of rolled and forged metallic components has been studied since the end of the nineteenth century. The main mechanisms of fatigue failure are described extensively in [2,3,4]. Plastic strain leads to clusters with higher dislocation densities than average [4]. Cyclic straining increases this heterogeneity [5]. After a sufficient number of cycles, cells or planar arrangements of dislocations are created. The slip is concentrated in thin persistent slip bands forming extrusions and intrusions on the surface [4]. Multiplication of extrusions and intrusions initiates a crack, which begins to propagate in shear mode (stage I of fatigue crack initiation). In stage II of fatigue crack initiation, the crack progressively aligns perpendicular to the load axis. The fatigue life corresponds to the number of cycles to initiate a dominant crack and to propagate the latter until catastrophic failure occurs. Crack initiation can be as high as some 90% of the total fatigue life [5]. Due to the variability of the initial flaw populations, fatigue results exhibit considerable scatter in the number of cycles to failure [2,3,4,5].

Three main loading domains may be distinguished in fatigue analyses [4]. Under giga-cycle (more than 10^6^ cycles) and high-cycle fatigue, the material deforms primarily elastically. The number of cycles to failure under such high-cycle fatigue has traditionally been characterized in terms of the stress range. The stresses associated with low-cycle fatigue cause appreciable plastic deformation prior to failure, and the fatigue life is characterized in terms of the strain range. A major application of low-cycle fatigue is the prediction of fatigue life in regions ahead of a stress concentration. The low-cycle approach to fatigue has found particularly extensive use in ground-vehicle industries [2].

Under low-cycle (high stress fatigue) the effect of small defects on the fatigue life is increased. Murakami [6] gives an extensive overview of the effects of small defects, nonmetallic inclusions, notches and cracks on the fatigue life of forged or rolled steels. Zerbst and Klinger [7] analyzed fatigue crack initiation by non-metallic inclusions, pores, micro-shrinkages and un-welded regions.

*Compared to rolled or forged work pieces, cast ones may contain shrinkage porosities* [1,8,9,10]. These porosities control to a large extend crack initiation (stages I, II). The formation of porosities during the solidification of steel is linked to two major phenomena [1,8,9,10]: the volume shrinkage of the steel and the release of gaseous elements. The first phenomenon gives rise to the formation of porosities called *micro-shrinkage*. A typical example of a micro-shrinkage porosity observed in G20Mn5 steel is shown in Figure 1. These porosities exhibit a large number of branches. Shrinkages are internal or external *macroscopic defects* resulting from the same phenomenon. Niyama presented one of the first criteria for shrinkage prediction [11]. Historically, acceptable classes of defects were defined. No distinction has been made between micro-shrinkages and sink marks [12,13,14,15]. Beckermann et al. [16] analyzed the effect of porosity size distribution on the fatigue behavior of 8630 cast steel. Serrano-Munoz analyzed the effect of close surface defects on the fatigue life [17]. Carlson and Beckermann suggested a dimensionless Niyama criterion [18], and Shouzhu et al. [19] suggested new metal feeding rules to prevent sink mark formation.

To the best of our knowledge, no detailed analyses of the influence of micro-shrinkages without the presence of macro porosities on low cycle fatigue damage of cast steels are available. Nevertheless, Han [20] highlighted that cracks initiate around those porosities in G20Mn5QT grade during high cycle fatigue (HCF) tests; as well as Wu in a G20Mn5N grade during low cycle fatigue (LCF) tests [21]. Moreover, Nagel studied the impact of internal and surface porosities inside cast specimens in G20Mn5Q grade during HCF tests [22]. Various types of porosities were produced thanks to two geometries of cast specimens, with or without a thicker central cross-section. It appeared that specimens cast with this larger shoulder contain a bigger central shrinkage, which reduces fatigue life. However, these kinds of porosities are rather macro-shrinkages linked to the hot spot in the middle of the specimen, which is not fed with liquid steel when it solidifies, than micro-shrinkages, which are formed according the same phenomenon between dendrites.

The present paper addresses the fatigue life of normalized G20Mn5 cast steel and the influence of the solidification conditions (columnar or equiaxed dendrites) on its microstructure, especially the size and the shape of micro-shrinkage porosities; then the influence of these micro-porosities on fatigue damage. This steel grade is the most produced by Safe Metal and commonly used for structural parts. Special ingots were produced with oversized risers and the absence of macro-porosity was controlled. The paper is organized as follows. The details of materials elaboration and experimental methods are given in Section 2. Section 3 is dedicated to the microstructural and monotonic mechanical characterizations. In Section 4, low-cycle fatigue test results are presented and correlated to the microstructure of the G20Mn5 steel.

## 2. Materials and Methods

### 2.1. Ingots Casting

Several ingots of G20Mn5 steel were cast specifically for this work, following the usual process of Castmetal Feurs foundry. The spray of the casting design (Figure 2) and the casting parameters were defined in order to avoid macro-shrinkages. Special care was taken to optimize feeding, ingots geometry and the use of the pencil core. The ingots composition (Table 1) was determined by spark emission spectrometry (Thermo Fisher ARL 4462, Walthamm, MA, USA). The analysis was conducted on a sample taken in the molted bath.

The ingots were separated from the riser by oxygen cutting. Ultrasonic inspection was conducted on each ingot. A normalizing heat treatment at 880°C followed by air cooling was applied.

### 2.2. Microstructure

On a first ingot, macroscopic etching was done on a full cross section prior to the normalization treatment. A 22% HCl solution at 70 °C was used during 20 min. As expected, the delimitation between columnar and equiaxed dendrites areas has been revealed (Figure 3a). Figure 3b,c show zooms of regions, respectively, in the columnar and equiaxed zone. On another ingot, after normalization, microstructural characterizations were conducted at different locations. Samples with a size of 20 × 10 × 10 mm were cut following the blueprint given in Figure 3d. Each sample was coated using compression mounting method in a Simplimet 1000 Buehler device. The samples were polished on an automatic Automet 250 Buehler device (Buehler, Leinfelden-Echterdingen, Germany) with SiC disks or diamond past, until mirror finish step. After polishing the samples, they were rinsed in ethanol, dried with compressed air and stored in a desiccator. The reactant used to reveal grain boundaries was Nital (5 HNO_3_/95 ethanol, vol.). Images were acquired with an optical microscope (Olympus GX 51, Evident Europe GmbH-French Branch, Rungis, French) at a magnification of ×200, then analyzed with a method defined in Olympus Stream software (https://www.olympus-ims.com/fr/microscope/stream2/ (accessed on 12 September 2022)).

A specific characterization procedure was developed to study porosities size, shape and localization in fatigue test pieces. X-ray Computed Tomography data were acquired with a General Electric Nanotom 180 micro-tomograph device (General Electric, Boston, MA, USA). The analyzed volume was a cylinder of 5 mm in diameter and 10 mm in length. The size of the voxel is 6.25 µm. The data were processed with Phoenix DatosX software (https://www.bakerhughes.com/waygate-technologies/ndt-software/phoenix-datosx-industrial-ct-scanning-software (accessed on 12 September 2022)). Porosity characteristics were then studied with Avizo software (version 2020). Considering the voxel size, only porosities containing at least 10 voxels, equivalent to a sphere with a diameter higher than 14 µm, are kept for further analyses.

### 2.3. Mechanical Tests

#### 2.3.1. Sample Preparation

Mechanical test specimens were milled from bars cut into two ingots by wire electro discharge machining. The positions of the bars in the ingots are given in Figure 4a. It is worth noting that bars are cut either on the border (red) or in the center (blue) of the ingots. Additionally, some bars have their axis along ingot length and other have their axis along ingot height. The same specimen geometry (Figure 4b) was used for monotonic tensile and low cycle fatigue tests.

#### 2.3.2. Monotonic Mechanical Tests

Tensile tests and compression tests were conducted on samples taken either close to the surface (red) or from the center of ingots (blue), in horizontal or in vertical direction (Figure 4a). The tensile test pieces geometry is given on Figure 4b. The initial gage diameter and gage length are, respectively, 5 mm and 10 mm. The tests were conducted on a Schenck RMC100 device controlled with Instron Console and Wavematrix. Compression test pieces were cylinders with a diameter of 10 mm and a height of 15 mm. The tests were conducted on a servohydraulic Schenck compression machine. The mechanical tests were strain rate or velocity controlled. The compression tests were done at a constant strain rate of 0.01 s^−1^. The monotonic tensile tests were conducted at constant cross-head velocity of 1 mm/s. Vickers hardness tests were also carried out on a Wolpert Testwell macrohardness under a load of 10 kgf and on a Matra MVK-1H microhardness testing machine under loads of 10 gf inside the ferrite and 25 gf inside ferrite and pearlite.

#### 2.3.3. Low Cycle Fatigue Tests

Low-cycle fatigue tests were conducted with an electro-mechanical Schenck RMC100 testing machine (Carl Schenck AG, Darmstadt, Germany). The test conditions are summarized in Table 2. Three levels of plastic strain range were defined in order to cover a wide range of low cycle fatigue. At each strain level, respectively, ten samples from the ingot skin and ten samples from the core were tested. In order to avoid damaging the fracture surface, a shutoff parameter was defined to stop every fatigue test when the maximal (tensile) stress (σ_max_) dropped more than 20% compared to a control stress (σ_cont._). The values of the control stress were measured during each test. For the lowest loading ($\Delta \varepsilon_{p}/2=0.02\%$), this value has been determined at cycle n° 500. For the two other loadings ($\Delta \varepsilon_{p}/2=0.1\%,\;0.4\%$), the value of the control stress (σ_cont._) was determined at cycle n° 100.

## 3. Microstructure and Mechanical Characterization

### 3.1. Microstructure

After the normalizing treatment, a ferritic–pearlitic microstructure was expected. Optical micrographs were acquired at different locations in the ingot, either in the center or in the skin part of the cross section. Figure 5 displays two typical examples of the ferritic–pearlitic microstructure observed, respectively, at the ingot skin an at the ingot center. The average grain size was measured on six samples taken from the ingot skin and on six samples taken from the core of the slab. The average equivalent diameter (Table 3) is almost the same from one sample to another. No significant difference was observed between samples taken from the core or the skin.

The populations of inclusions were characterized prior to etching. In accordance with the chemical composition of the G20Mn5 steel grade, mainly manganese sulfides (MnS) especially MnS type III (Figure 6) were observed. The composition was checked by EDS analysis. Based on [23] indexes of inclusion cleanliness were determined (Table 4). For both fine and thick series (corresponding, respectively, to inclusions with a thickness below or above 8 µm), indexes are very close between the skin and the core of the ingot.

Special attention was given to porosities characterization. Three-dimensional X-ray tomography observations were made on several fatigue samples prior to mechanical testing. A significant difference between samples taken from the core and those taken from the skin of the ingot appears. Table 5 summarizes the mean values of the porosity distributions. The mean void volume fraction is slightly larger in the skin of the ingot than in the core. However, the main difference concerns the void volume distribution. The number of voids per unit volume is much smaller in the samples taken from the core of the ingot. The mean micro-shrinkage volume is much larger in the core than in the skin of the ingot. Figure 7 depicts the probability density as function of the void (micro-shrinkage) volume for the core and skin of the ingot. The skin of the ingot is characterized by a rather sharp peak, whereas in the core of the ingot very large voids (0.003 mm^3^) are observed. The core contains thus a smaller amount of micro-shrinkages than the skin; but much higher volumes are observed.

The probability distributions of voids in the core and skin are different. The mechanical effect of voids depends on the size, but also on the shape and location in the sample. The shape of voids is often characterized by their sphericity. Figure 8a depicts the sphericity as a function of the void volume. Obviously, the sphericity decreases strongly when the void volume is increased. Thus, only small voids exhibit a quasi-spherical shape. For voids with a volume larger than v_c_ = 0.00025 mm^3^, the sphericity is smaller than 0.4 (red dashed lines in Figure 8a). For both locations (ingot skin or ingot core), voids with volumes larger v_c_ are present. Among all platonic solids, the tetrahedron exhibits the smallest sphericity (0.671). As, in present work, the voids are associated with micro-shrinkages, the description depicted in Figure 8b seems more adequate. The total surface of the object (sum all external surfaces) was determined. Figure 8b shows the total void surface as a function of the void volume. For both locations (skin or core), a perfectly linear relationship between the void surface and the void volume is observed. This linear relation implies quasi-two-dimensional void shapes as represented schematically in Figure 8b. If the “thickness” h is always small compared to the in plane dimensions (d_1_,d_2_) a linear relationship between the void surface and void volume is satisfied, even if the in-plane shape is much more complex in reality. The shapes shown in Figure 8b satisfy the condition of a constant ratio volume/total surface and allow small values of the sphericity parameter. Indeed, as an example, we assume d_1_ = d_2_. Then, a sphericity of 0.4 leads to h/d_1_ = 0.085.

The microstructural observations may be summarized as follows. After the normalizing treatment, a ferritic–pearlitic microstructure was observed at the ingot skin and in the ingot center. The average grain size is almost the same in the skin and the core of the ingot. The inclusion distribution in the skin and the core of the ingot are very close. The main difference between the ingot skin and core corresponds to the presence of larger micro-shrinkages in the core than in the skin of the ingot.

### 3.2. Monotonic Mechanical Properties

Vickers macro-hardness and micro-hardness tests were conducted at different locations in the skin and core of the ingot. For macro-hardness, 12 indents were done on each part. Microhardness indents were located either in ferrite or in pearlite. For each phase, 30 indents and 40 indents were done in the skin parts and core part, respectively. The results are given in Table 6. The difference in hardness between the ingot skin and core is contained in the error bars. The hardness of the skin and core are thus considered identical. The same conclusion holds for the micro-hardnesses in both phases.

Tensile and compression tests were conducted on samples from the ingot skin and core. Monotonic test results are presented in Figure 9. Monotonic tensile and compression behavior for the ingot skin and core are identical. The tensile curves exhibit the Portevin-Le Chatelier effect and Lüders bands typical of low-carbon steels. Very high strain hardening at small strains (<5%) is observed. Comparison of tensile and compression curves hints that significant damage in tenson occurs after 5% (point A). However, macroscopic necking occurs much later at ~18% (point B). The significant effect of damage at low stress triaxiality (prior to necking) corresponds to the large initial void volume fractions. However, the large strain hardening (compression curves) leads to a significant ultimate strength of about 520 MPa and a large ductility of 37%. The monotonic test results are summarized in Table 6.

## 4. Low Cycle Fatigue Tests

Low cycle fatigue tests, with $R_{\epsilon }^{p}=-1$, were conducted for three levels of plastic strain (Δε^P^/2 = 0.02%, 0.1% and 0.4%, respectively) on 10 samples from the ingot core and 10 samples of the ingot skin. First, a control value of the stress ($\sigma_{cont.}$) was *determined during* the test. The tests were stopped when the tensile stress became smaller than 0.8$\sigma_{cont.}$. From now on, the number of cycles achieved precisely at $\sigma_{max}=0.8\sigma_{cont.}$. will be called N_f_. The final rupture was obtained by simple tension. This is the classical way to preserve a fracture surface with $R_{\epsilon }^{p}=-1$.

### 4.1. Low Cycle Fatigue S-N-Curves

Figure 10 shows the corresponding results. In each column, the plastic strain increases from top to bottom. In each row samples from the ingot core and skin are shown. Comparing the results corresponding to different plastic strain levels (column) shows the sharp decrease in the number of cycles for increasing plastic strain for samples from the ingot core as well as samples from the skin.

A significant dispersion of the stress has to be noted. At the very beginning of the tests, dispersion may be attributed to differences in the microstructure, but also to small misorientations of the samples. Hence, vertical dashed lines delimit on each graph a very small portion of the S-N-curves, which will not be considered in the interpretation. The following discussion concerns only the parts to the right of the red dashed lines. Even to the right of the dashed lines, significant dispersion is observed. This dispersion decreases if the applied plastic strain and hence the applied stress increase. At the lowest level of plastic strain (0.02%) during accommodation, a decrease of the stress is observed. At the intermediate strain increment, no significant softening nor hardening occurs. Finally, at the highest plastic strain level (0.4%), significant strain hardening occurs. In Section 4.2, the number of cycles will be analyzed by Manson Coffin’s approach. In Section 4.3, a damage analysis will allow to distinguish between subcritical and final crack growth.

### 4.2. Number of Cycles at Failure

#### 4.2.1. Manson–Coffin Parameters

The results presented in Figure 10 allow determining the mean number of cycles at failure for all the considered samples (Table 7). For all the considered plastic strains, samples from the ingot skin exhibit an average number of cycles at failure (N_f_) larger than specimens from the core.

Under low cycle fatigue conditions, Manson and Coffin [24,25] postulated a simple power–law relation between the numbers of cycles at failure (N_f_) and the plastic strain level ($\Delta \varepsilon_{P}$).
(1)$$\frac{\Delta \varepsilon_{P}}{2}=\varepsilon_{f}^{\prime }\;N_{f}^{c}$$$\varepsilon_{f}^{\prime }$ and c are material parameters. Based on the results of Table 7, the Manson–Coffin parameters for skin and core samples taken separately and for all samples were determined (Table 8).

Figure 11 shows a graphical representation of the plastic strain increment as a function of the number of cycles at failure. This representation leads to a small difference between the behaviors of samples from the ingot skin and ingot core. The mean behavior (skin+ core samples) is represented in solid green (MC). The global behavior is very close to the one determined by Han [20] for G20Mn5QT cast steel at R = −1 (green dashed line). However, a significant difference with the curve from Wu’s study [21] who characterized the material at R = 0.1 (dotted green line). This difference hints at a significant influence of the average stress on the fatigue life. Finally, looking at the total strain by adding Basquin’s law (yellow curve) allows us to match HFC results obtained in a previous study.

#### 4.2.2. Difference between Ingot Core and Skin

The number of cycles at failure based on the combination of skin and core samples gives only a first indication of the material behavior under cyclic loading and allows to show good correspondence with Han’s results. A more detailed analysis of the number of cycles at failure is given in Table 9. The mean values correspond to Table 7. The main focus is now on the dispersion of the results (standard deviation/mean value). Results corresponding to the ingot core exhibit always a larger dispersion than samples from the ingot skin. For both series the dispersion increases with the applied plastic strain. For each applied plastic strain, the number of cycles at failure (N_f_) between samples from the skin and the core of the ingot differs significantly (15% to 51%). However, the stabilized stress exhibits almost the same value between both series.

### 4.3. Remaining Load Carrying Capacity during Cycling

The number of cycles at failure gives a first precious indication of the fatigue life expectancy for a fixed load level (Δε^P^/2). However, for different successive loadings, the remaining load carrying capacity of the material after a given number of cycles is significant. Figure 12 depicts the stress as a function of the normalized number of cycles (N/N_f_) for all considered loadings.

Remarkably, for all considered strain levels and both sample locations, the sharp stress decrease occurs after about 90% of the fatigue life time (red dashed line AA’). This corresponds to the transition from subcritical to critical crack growth at a number of cycles N_c_. In the following paragraph, a method to exactly determine N_c_ is elaborated. The number of cycles at transition N_c_ can be determined. The crack length at N_c_ is not determined. This method is based on damage analysis. A damage variable *D* may be defined based on the stress response [26]. The most simple definition of the damage variable assuming isotropic material behavior is as follows:(2)$$\tilde{\sigma }=\sigma \left(1-D\right)$$$\tilde{\sigma }$ and σ designate, respectively, the apparent stress in the damaged material and the corresponding stress in the sound material. Under monotonic loading, the apparent stress $\tilde{\sigma }$ varies from σ (sound material) to zero (*D* = 1) at failure. Under fatigue loading, the cycling is stopped prior to the final failure and the samples are broken under simple tension. Nevertheless, the evolution of the damage parameter in the function of the normalized number of cycles N/Nf gives precious indications on the remaining load carrying capacity. Lemaitre and Chaboche [26] suggested different damage variables *D*.

Three different damage variables were determined in present work. The first *D*(σ) compares the stress at a given cycle, to the stress after stabilization σ_stab_.
(3)$$D\left(\sigma \right)=1-\frac{\sigma \left(N\right)}{\sigma_{stab}}$$
where $\sigma \left(N\right)$ and $\sigma_{stab}$ correspond, respectively, to the reached tensile stress at cycle *N* and at the stabilized cycle. The second damage measure is based on the ratio between the apparent Young’s modulus and the value of Young’s modulus at stabilization:(4)$$D\;\left(E\right)=1-\frac{E\left(N\right)}{E_{stab}}$$
where $E\left(N\right)$ and $E_{stab}$ correspond, respectively, to the Yound modulus measured at a given cycle and at the stabilized cycle. The last damage measure considered in present work is based on the difference between the tensile and the compressive stresses at each cycle normalized by the stress after stabilization.
(5)$$D\left(TC\right)=1-\left|\frac{\sigma_{T}\left(N\right)-\sigma_{C}\left(N\right)}{\sigma_{stab}}\right|$$
where $\sigma_{T}\left(N\right)$ and $\sigma_{C}\left(N\right)$ correspond, respectively, to the compressive stresses reached at a given cycle and at the stabilized cycle.

These most common damage variables are depicted on Figure 13 as a function of the normalized number of cycles N/N_f_ for the three considered plastic strain levels. Negative values correspond to initial softening. The damage varies slowly up to a value N/N_f_ ~0.9. Then, a very steep increase follows.

At a given plastic strain, the three damage variables indicate the same transition point from slow variation to steep increase. Any of the three common damage variables may thus be used to determine transition from damage initiation (sub-critical growth) to macroscopic growth. The combined knowledge of the numbers of cycles at failure (N_f_) and the transition from the initiation stage to accelerated growth gives precious results for determining the remaining load carrying capacity at a given number of cycles. As the three common damage variables D(E), D(TC) and D(σ) lead to similar transition points, we suggest working with D(σ). As the current stress at each cycle is measured for all common fatigue experiments, D(σ) is known.

To determine the transition from initiation to rapid growth, both regimes have been represented by straight lines as suggested in [26]. The intercept between both lines determines the critical number of cycles N_c_ at the transition. Figure 14 depicts the corresponding results for all considered plastic strains and both sample locations. For all considered load levels and locations, the transition between both regimes occurs at N_c_/N_f_ > 92%. The initiation regime is thus always larger than 90% of the sample lifetime. Remarkably, transition occurs at the smallest ratio N_c_/N_f_ for the core sample loaded at $\Delta \epsilon^{p}/2=0.4\%$. The combination of large applied remote stress and the of large micro-shrinkages leads to a reduced lifetime (N_f_) and a reduced initiation state (N_c_/N_f_).

### 4.4. Microscopic Analysis of Damage and Murakami Type Law

#### 4.4.1. Different Damage Parameters

Let us briefly summarize the previous results. The ferritic–pearlitic microstructure observed after normalizing treatment exhibits the same grain size and same inclusion distribution in the skin and the core of the ingot. The monotonic material behavior (stress–strain curve, hardness and micro-hardness in both phases) of samples from the skin and core of the ingot are very close. The sole significant differences between skin and core samples are the presence of larger micro-shrinkages in the core than in the skin, and a reduced number of cycles at failure (N_f_) as well as a reduced initiation state (N_c_/N_f_) for samples from the core compared to samples from ingot skin. This hints to a close relation between the initial porosity and the number of cycles at failure (N_f_). Moreover, the micro-shrinkages exhibit a strictly linear relationship between the volume and the surface. Defects exhibiting a large volume under tomographic observation should thus give rise on the sample fracture surface to defects with large areas.

The fracture surfaces of all samples were characterized by SEM-observations. These observations confirmed that initiation occurs always at micro-shrinkage porosities. In samples from the ingot core, cracking starts in most cases (80%) at a single large micro-shrinkage porosity. In samples from the ingot skin, initiation at single and at multiple shrinkage porosities were observed. In the present work, only examples with single crack initiation are shown. We found that multiple crack initiation was favored if the plastic strain was increased (Δε_P_ = 0.4%,). If the plastic strain is increased, the remote stress increases as well, and thus more initial defects will correspond to a critical size.

Murakami et al. [27] closely analyzed the relationship between fatigue life time and initial defect area. The relevance of several damage variables based on this concept is analyzed. Figure 15 depicts a typical fracture surface and explains the measurements made to determine different damage variables.

The cross-sectional area *S*_0_ was determined on each sample. The initial defect has an area $A_{d}$ and the crack an area $A_{c}$. These areas were measured as the total area of the corresponding pixels. Equivalent ellipses were defined on the initial defect (minor axis $m_{d}$) and the fatigue crack (minor axis $m_{c}$). The most commonly used damage parameters are defined as follows:(6)$$D_{1}=\frac{A_{c}-A_{d}}{S_{0}}\;\;\;D_{2}=\frac{A_{c}^{eq}-A_{d}^{eq}}{S_{0}}\;\;\;D_{3}=\frac{m_{c}-\sqrt{A_{d}}}{\varphi_{0}}\;\;\;D_{4}=\frac{m_{c}-m_{d}}{\varphi_{0}}$$$A_{c}^{eq}$ and $A_{d}^{eq}$ designate, respectively the areas of both equivalent ellipses.
(7)$$A_{c}^{eq}=\frac{\pi M_{c}m_{c}}{4}\;\;\;\;\;\;\;\;A_{d}^{eq}=\frac{\pi M_{d}m_{d}}{4}$$*D*_1_ compares the areas of the fatigue crack and the initial defect. *D*_2_ corresponds to a similar measure based on the equivalent ellipses. *D*_3_ favors the propagation direction of the crack and considers an isotropic influence of the initial defect. Finally, in measure *D*_4_ the main weight corresponds to the crack’s and initial defect’s dimensions in the propagation direction.

For all samples exhibiting a crack initiated at a single defect, the values of the four damage variables have been computed. Table 10 gives the mean values for the three considered strain levels. All damage parameters lead to higher values in the ingot core than the ingot skin. For both locations, the values of all damage variables increase with increasing plastic strain. The two damage parameters based on area measurements (*D*_1_ and *D*_2_) exhibit exactly the same values. These two damage parameters lead to smaller values than the two damage parameters (*D*_3_ and *D*_4_) based on length measurements in the propagation direction. Note the microscopic measurements of damage lead to larger values than the shut-off parameter based on the load in tension.

#### 4.4.2. Murakami Type Analysis

Based on the previous results, the correlation between the number of cycles at failure with a stress intensity factor based on the initial defect has been tested. *The number of cycles N_f_ may not be related directly to any stress intensity factor*. However, the number of cycles corresponding to the final stage (stage II), i.e., (N_f_-N_c_)/N_f_, may be related to the stress intensity factor based on $\sqrt{A_{d}}$ and the stabalized stress $\sigma_{stab}$. Xua et al. [28] successfully modelled fatigue crack propagation (stage II) in G20Mn5QT cast steel by a two parameter model based on a Smith–Watson–Topper approach. The authors introduced a material correction to the classical SWT-parameter. An overview of the different SWT-parameters used in the literature is given in Łagoda et al. [29]. A similar approach is not immediate when considering stage I followed by stage II. In the present work, only the initial defect and the final crack shape could be observed by SEM characterization. Thus, the following classical expression for stress intensity factor *K*, based on Murakami’s work, was adopted:(8)$$K=\sigma_{stab}.\;\sqrt{\pi \sqrt{A_{d}}}$$

Figure 16 depicts the stress intensity factor *K* as a function of the normalized number of cycles in stage II. For Figure 16a the classical logarithmic axes were adopted, whereas Figure 16b shows the same result in cartesian axes. A satisfying linear relationship in the log–log (Figure 16a) was obtained for core samples for all the considered plastic strain increments. For the skin samples, no linear relationship is observed. As only samples with a single initial defect were considered, this might be due to the reduced number of samples. However, the dispersion increases drastically with the plastic strain Δεp or the corresponding stabilized stress $\sigma_{stab}$. Under large plastic strain, the increased number of “smaller” defects in the skin sample leads to an increased number of defects almost perpendicular to the remote stress. For (∆εp/2 < 0.4%), these defects are too small to be activated. In the core samples, this dispersion is not seen, because larger defects (reduced in number) have been activated.

## 5. Discussion and Conclusions

The low cycle fatigue behavior of G20Mn5 cast steel was analyzed. Particular castings with large risers were used to eliminate all macro-shrinkages. First, the columnar and equiaxed zones were determined on an as cast ingot. Then, only normalized samples were analyzed. Monotonic tensile and compression curves show that microvoids have no influence on the stress strain curve up to 5%.

The distinction between **C**ore and **S**kin samples was made based on macrographic observations of the as-cast microstructure. Then, the normalized steel was characterized thoroughly. First, the micro-shrinkage-porosity was characterized on all LCF samples by Micro-CT-scans. Low cycle fatigue tests on normalized G20Mn5 cast steel were conducted at three levels of plastic strain (0.02%, 0.1% and 0.4%).

The main results may be summarized as follows. The microstructure and monotonic mechanical behavior of the ingot skin and core may be considered identical. The sole difference between the core and the skin is controlled by the micro-shrinkage distribution. The core exhibits a widespread volume distribution and the skin one much narrower. However, both core and ingot contain essentially flat (quasi two-dimensional) micro-shrinkages. At this stage, all differences in fatigue life between samples from the ingot skin or ingot core may be attributed to differences of the micro-shrinkage volume distributions.

For all fixed plastic strain increments and both locations (Skin or Core) similar curves are obtained. Depending on the plastic strain, strain hardening or softening may be observed. The stress stabilizes (σ_stab_) at about 10^2^ cycles. A Manson–Coffin description with parameters from the literature allows prediction of the numbers of cycles at failure N_f_. The number of cycles at failure is always smaller in the core than ingot skin.

Introduction of the simple damage variable (D = σ/σ_stab_), based on values recorded during cycling, shows that the initiation stage (N < N_c_) spans over more than 90% of the lifetime. The total number of cycles at failure N_f_ and the number of cycles corresponding to the initiation stage (N_c_) are smaller in the ingot core than in its skin. This naturally led to a damage characterization based on the observation of the fracture surfaces.

All fracture surfaces were characterized by FEG-SEM observations. All fatigue cracks have been initiated at porosities corresponding to micro-shrinkages. On samples with a single micro-shrinkage responsible for crack initiation, several damage parameters were defined. The most discriminant are based on the characteristic lengths of the initial defect and fatigue crack in the crack propagation direction. Based on these results, a Murakami damage type analysis was done. The stress intensity factor calculated with the initial defect size is related to the number of cycles in the final stage (N_f_-N_c_)/N_f_.

## Figures and Tables

**Figure 1 materials-15-07072-f001:**
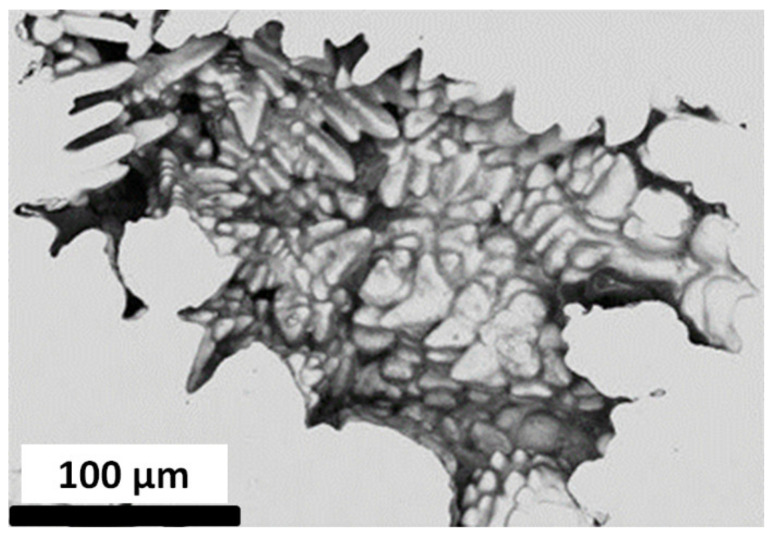
Micro-shrinkage porosity.

**Figure 2 materials-15-07072-f002:**
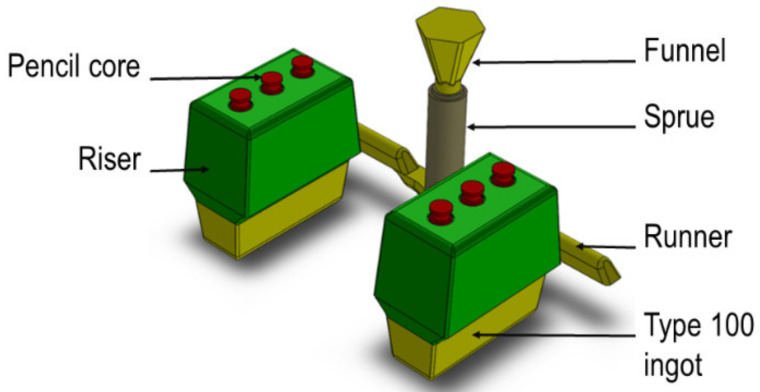
Casting device.

**Figure 3 materials-15-07072-f003:**
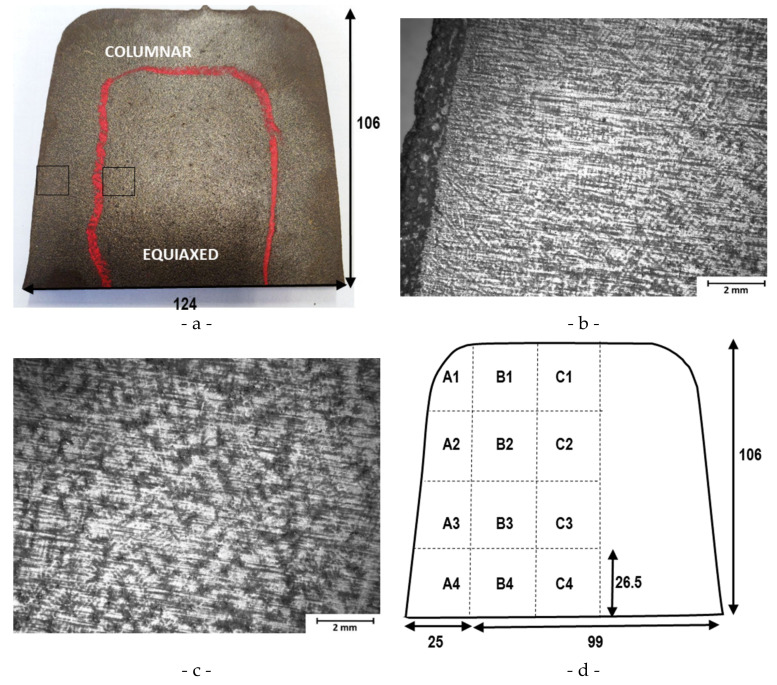
(**a**) Ingot cross section prior to normalization treatment. Red: boundary between columnar and equiaxed the black squares correspond to the zooms presented in figures (**b**,**c**). (**b**) Columnar zone, (**c**) equiaxed zone. (**d**) Sample cutting localization and labels for microstructure analyzes on an ingot after normalizing heat treatment.

**Figure 4 materials-15-07072-f004:**
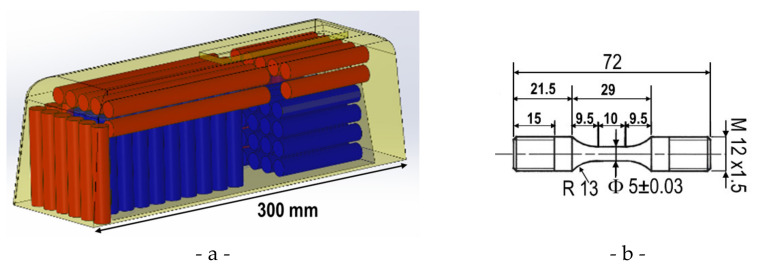
(**a**) Type 100 ingot with indication of wire cutting positions (skin in red, core in blue) for cylindrical bars. (**b**) Tensile and fatigue sample geometry.

**Figure 5 materials-15-07072-f005:**
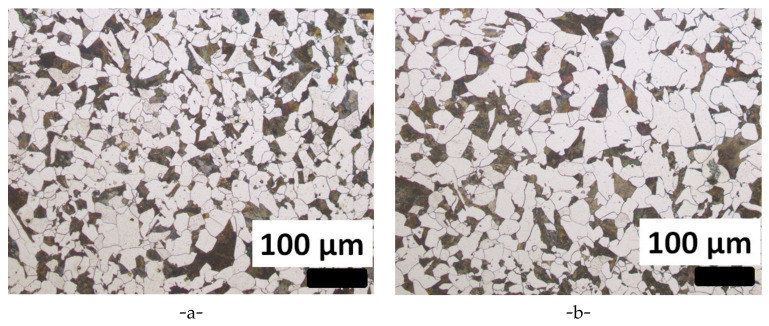
Metallographic observations after Nital etching in: (**a**) the skin (A4) and (**b**) core (C4) parts.

**Figure 6 materials-15-07072-f006:**
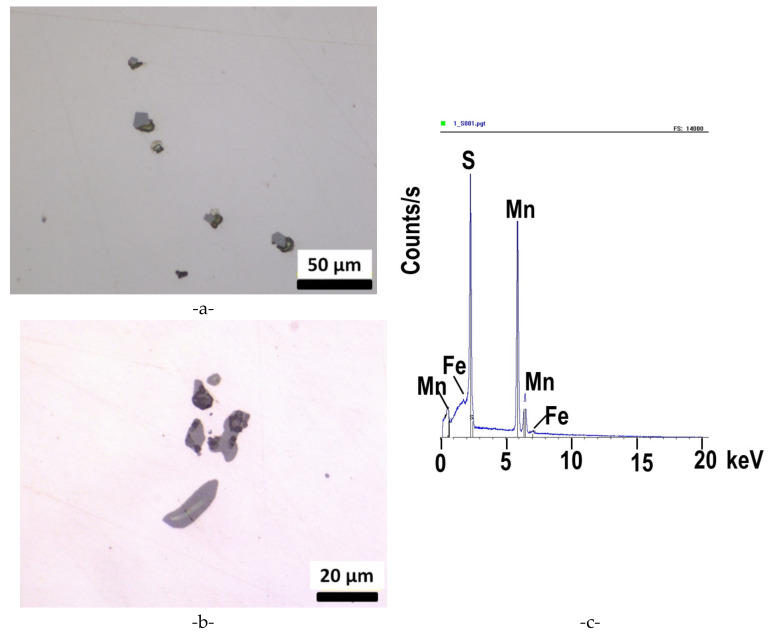
MnS type III (**a**,**b**), typical EDS observation of one of the particles (**c**).

**Figure 7 materials-15-07072-f007:**
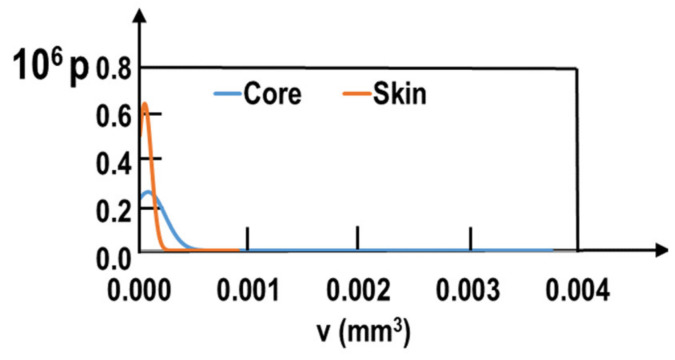
Probability density distribution p in function of the void volume v.

**Figure 8 materials-15-07072-f008:**
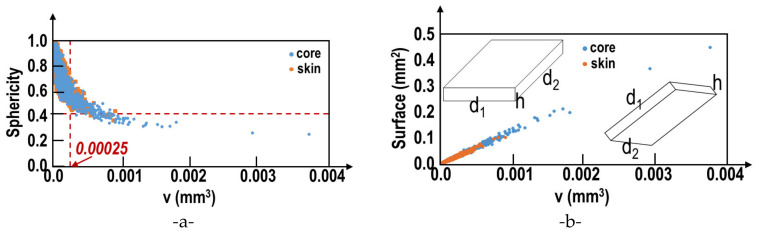
Micro-shrinkage characterization. (**a**) Sphericity in function of the volume v, and (**b**) total surface in function of the volume v.

**Figure 9 materials-15-07072-f009:**
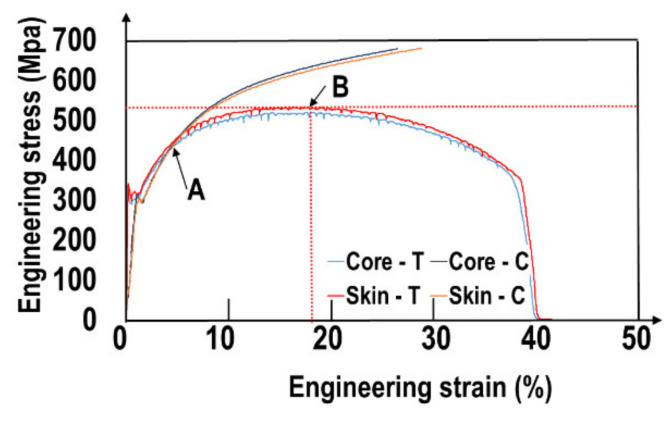
Tensile and compressive curve for skin and core specimens.

**Figure 10 materials-15-07072-f010:**
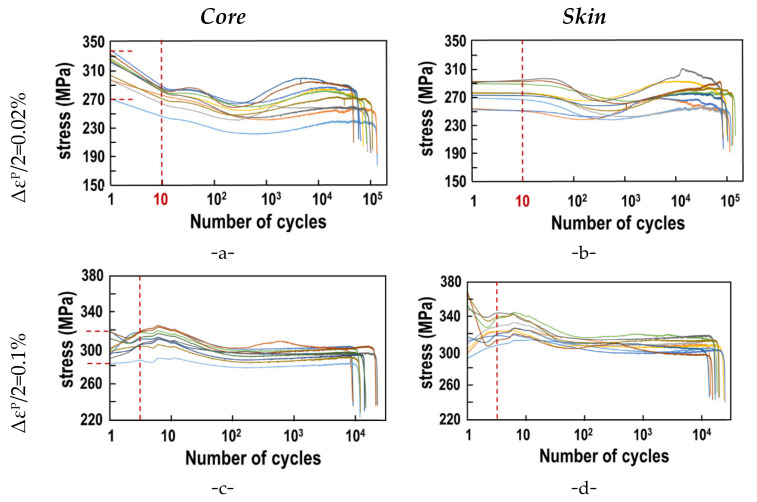
Fatigue test results (stress vs. number of cycles) for different values of plastic strain Δε^P^/2 = 0.1% (**a**,**b**); 0.2% (**c**,**d**) and 0.4% (**e**,**f**). Samples for the ingot core are represented in (**a**,**c**,**e**) and from the ingot skin in (**b**,**d**,**f**).

**Figure 11 materials-15-07072-f011:**
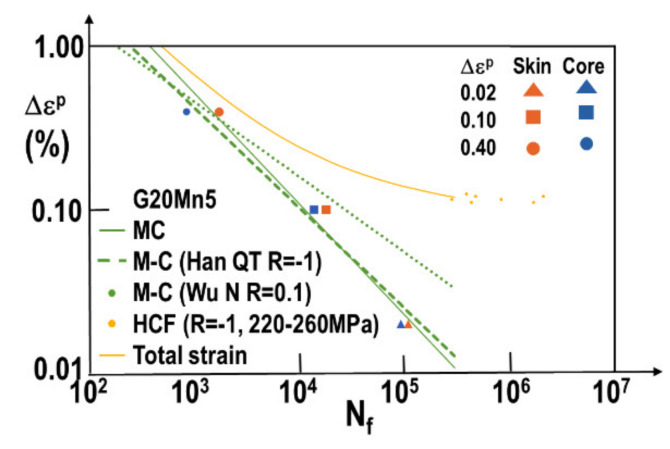
Manson–Coffin plot based on experimental values obtained in this work and compared to literature data. The latter are Han et al. work on quenched G20Mn5 and Wu et al. work on normalized G20Mn5. HCF data (yellow points) were obtained with applied stress tests. Corresponding elastic strain values are used to plot the results on this strain vs. number of cycles to rupture graph.

**Figure 12 materials-15-07072-f012:**
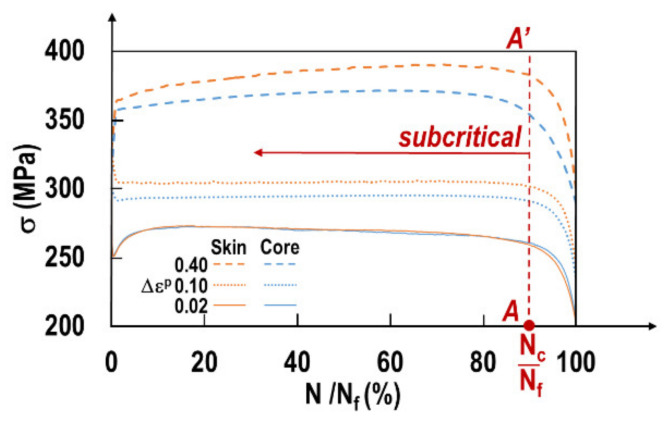
Stress as function of the normalized number of cycles for all considered loadings.

**Figure 13 materials-15-07072-f013:**
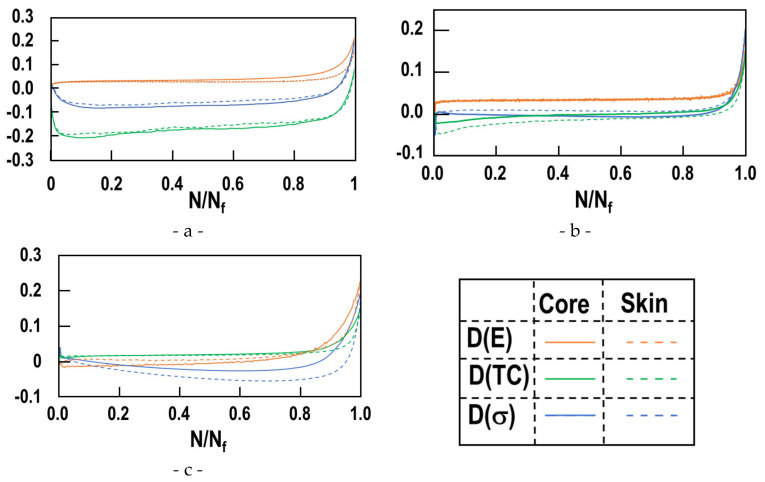
Comparison of three most common damage variables for all considered strain levels Δε^P^/2: 0.01% (**a**), 0.2% (**b**), and 0.4%(**c**).

**Figure 14 materials-15-07072-f014:**
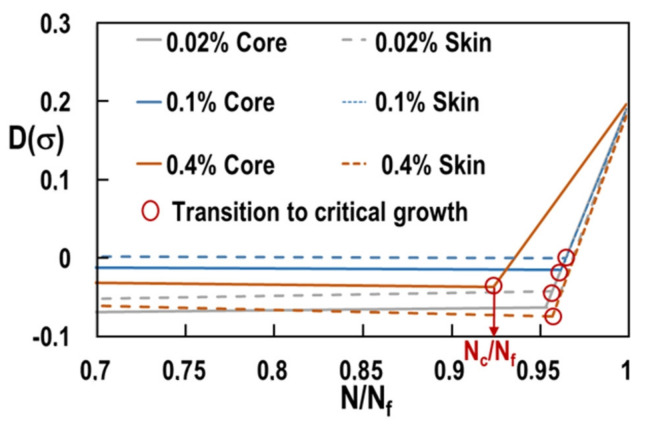
Damage D(σ) as a function of the normalized number of cycles. Determination of the critical number of cycles NC corresponding to the transition from sub-critical to critical growth.

**Figure 15 materials-15-07072-f015:**
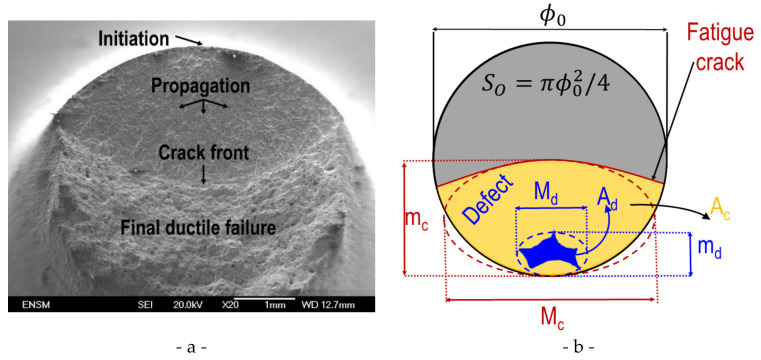
Fracture surface analysis. (**a**) Typical fracture surface with crack initiation at a single defect. (**b**) Different measurements made on each fracture surface to determine the values of the corresponding damage variables.

**Figure 16 materials-15-07072-f016:**
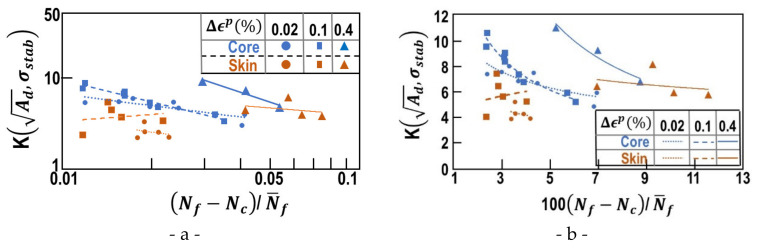
Correlation between stress intensity factor based on the initial defect area A_d_ and the normalized number of cycles in stage II. (**a**) Classical representation with (**b**) the same results in cartesian axes.

**Table 1 materials-15-07072-t001:** G20Mn5 chemical composition (%wt).

C	Mn	Si	S	P	Ni	Cr	Mo	V	Al	Fe
0.18	1.12	0.36	0.009	0.011	0.07	0.26	0.05	0.005	0.05	97.8

**Table 2 materials-15-07072-t002:** Low-cycle fatigue test parameters. Three different levels of plastic deformation. The plastic load ratio ($\epsilon_{p}^{min}/\epsilon_{p}^{max}$) is equal to −1.

Strain Level	$\frac{\Delta \varepsilon_{p}}{2}\;(\%)$	Signal (Strain)	Censoring (Cycles)	Shutoff
1	0.02	Triangular	10^6^	$\sigma_{max}\leq 0.8\;\sigma_{cont.}$
2	0.1		10^5^	
3	0.4		10^4^	

**Table 3 materials-15-07072-t003:** Mean grain size in the skin and in the core of the ingot. $\bar{d}$ designates the average equivalent grain diameter.

Location	$\bar{d}\;(µm)$	G Index
Skin	16.4	8.8
Core	16.9	8.9

**Table 4 materials-15-07072-t004:** Indexes of inclusion cleanliness.

Location	MnS Type III (Fine)	MnS Type III (Thick)
Skin	1.13	0.60
Core	1.08	0.68

**Table 5 materials-15-07072-t005:** Mean values of the porosity distribution in the skin and the core of the ingot. $\bar{f}$ and $\bar{N}$ correspond, respectively, to the mean void volume fraction and number of voids per unit volume. $\bar{v}\;$ and $\bar{d}\;$ designate, respectively, the mean void (micro-shrinkage) volume and diameter.

Location	$\bar{f}\;(\%)$	$\bar{N}$ $\left(mm^{-3}\right)$	$\bar{v}\;(µm^{3})$	$\bar{d}\;(µm)$
skin	0.0104	403	5.3 10^4^	46.6
core	0.00947	234	8.8 10^4^	55.2

**Table 6 materials-15-07072-t006:** Monotonic test results. Average values from three tests for tensile tests and tens of tests for hardness in each batch (core or skin).

	Core	Skin
UTS (Mpa)	518	528
Rp_0.2_ (Mpa)	293	309
A% (%)	37%	37%
Ferrite micro-hardness (HV)	113	121
Pearlite micro-hardness (HV)	303	291
Macro-hardness (HV)	140	142

**Table 7 materials-15-07072-t007:** Number of cycles at failure (Nf) as function of the plastic strain.

Δε^P^/2					
0.02%		0.1%		0.4%	
Core	Skin	Core	Skin	Core	Skin
92339	108684	13796	17940	835	1697

**Table 8 materials-15-07072-t008:** Manson–Coffin coefficients.

	$\varepsilon_{f}^{\prime }$	c
Skin	0.2996	0.714
Core	0.8822	0.626
Global	0.5536	0.677

**Table 9 materials-15-07072-t009:** Detailed results from low cycle fatigue tests.

		Δε^P^/2					
		0.02%		0.1%		0.4%	
		Core	Skin	Core	Skin	Core	Skin
Number of cycles at failure N_f_	mean	92339	108684	13796	17940	835	1 697
	stand. dev.	29511	20524	4703	3776	433	704
	stand. dev./mean	34%	20%	34%	21%	52%	41%
	max.	139875	148455	22677	24428	1741	2617
	Min.	48754	83775	8937	13195	473	298
$\frac{\left(N_{f}^{skin}-N_{f}^{core}\right)}{N_{f}^{skin}}$		**15%**		**30%**		**51%**	
σ_stab_. (MPa)		251.5	251.7	292.2	308.6	362.2	369.4

**Table 10 materials-15-07072-t010:** Damage measurements on fatigue specimens fracture surfaces.

Damage Variable	Δεp/2					
	0.02%		0.1%		0.4%	
	Core	Skin	Core	Skin	Core	Skin
*D* _1_	28%	24%	30%	29%	30%	35%
*D* _2_	28%	24%	28%	29%	30%	35%
*D* _3_	34%	33%	36%	38%	39%	44%
*D* _4_	35%	34%	36%	38%	38%	43%

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
