# Peer review of "Low Cycle Fatigue of G20Mn5 Cast Steel Relation between Microstructure and Fatigue Life"

_materials, 2022, doi:10.3390/ma15207072_

Round 1

Reviewer 1 Report

(1) Please check the number of references in introduction section. The reference [4] appeared in lines 39, 42, 49 ?  The references 10-12 were not listed in introduction?

(2) Some new references are encouraged to add in introduction, especially for the latest three years references.

(3) Error! Reference source not found appeared in lines 99, 102, 117, 119, 173, 184, 239 ?

(4) The loading criterion for monotonic tensile and Low Cycle Fatigue tests should be added in section 2.3.

(5) The quality of figures should be improved. 

Author Response

Thank you for your review.

Reviewer 2 Report

Very fine work. Just a few minors mentioned below

Figure 3 is not needed

Figure 4(a) difficult to see the details

Figure 7 needs EDS to confirm

Figure 17 difficult to see

5 Conclusions

Author Response

Thank you for your review.

Reviewer 3 Report

The article highlights peculiarities of low cycle fatigue behavior of G20Mn5 cast steel. The authors have evaluated distinction between Core and Skin samples. However, they paid insufficient attention to the investigation of the microstructural features of each of the studied zones (Core and Skin).

 The article is interesting, but a number of shortcomings need to be corrected:

1.     It is impossible to conduct a qualitative review if there are no references to Figures 1-4, 6, 7, 10, 16 and Table 1.

2.     In the presented figures, it is impossible to clearly identify the delimitation between columnar and equiaxed dendrites areas. A higher magnification microstructure should be provided.

3.     The caption to Fig. 9 does not match the figure (Fig. 9a Total surface in function of the volume v???; Fig. 9b sphericity in function of the volume v???).

4.     The values of engineering strain are unclear (very small strain values at A=37%). It is difficult to know exactly where the 5% and 18% deformation described in Lines 242-244 will be.

5.     The statement The tests were stopped when the tensile test was smaller than 0.8?????” (Line 257) is unclear. Maybe the authors meant “The tests were stopped when the tensile stress was smaller than 0.8?????”?

6.     For a better analysis, the ordinate axes in Fig. 11 for Core and Skin should be made the same for the same kind of tests (De).

7.     The authors should explain how they distinguish between crack initiation and crack growth (Lines 269-271). They should also explain on the basis of which results they believe that strain hardening occurs during subcritical crack growth.

8.     The authors should explain why for Core the stresses at the lowest level of plastic deformation (0.02%) are greater than at deformations (0.1%) according to Fig. 11a compared to Fig. 11c. This is unconventional.

9.     The authors should clearly explain whether the samples were fractured during the fatigue tests (Fig. 11) or not, since it is impossible to draw an unambiguous conclusion from Lines 257-258.

10.  The authors state that The sole significant differences between skin and core samples are the presence of larger micro-shrinkages in the core than in the skin…” (Lines 386-387). However, not only the size of micro-shrinkages is important, but also the geometric parameters (sphericity) of micro-shrinkages. This parameter (Fig. 9a) should be given more attention.

11.  The authors should explain how the initiation state (Nc/Nf) was determined, in particular, how Nc was determined. What was the initial size of the crack?

12.  The authors should provide confirmation of the statements made in Lines 394-397, since Fig. 16a is not convincing enough.

13.  Fig. 17 is of low quality, and the characters cannot be recognized.

14.  More new References (2019-2022) should be added.

Author Response

Thank you for your review.

Round 2

Reviewer 1 Report

The paper had been revised as suggestions

Author Response

Thank you so much for all the time spent on this publication.

Reviewer 3 Report

The authors took into account the comments of the reviewer and made appropriate corrections to the manuscript. 

But the text of the article needs to be corrected (Line 105, 108, 123, 126, 191 "Error! Reference source not found." should be replaced; Line 403 - figure number should be 13)

Author Response

Reviewer 3

(1)            The paper had been revised as suggestions.  The authors took into account the comments of the reviewer and made appropriate corrections to the manuscript.

But the text of the article needs to be corrected (Line 105, 108, 123, 126, 191 "Error! Reference source not found." should be replaced; Line 403 - figure number should be 13)

Thank you so much for all the time spent on this publication.

The following corrections were made:

Line 105 :

Several ingots of G20Mn5 steel were cast specifically for this work, following the usual process of Castmetal Feurs foundry. The spray of the casting design (Figure 2) and the casting parameters were defined in order to avoid macro-shrinkages.

Line 106:

The ingots composition (Table 1.) was determined by spark emission spectrometry…

Lines 117-120

As expected, the delimitation between columnar and equiaxed dendrites areas has been revealed (Figure3 a). Figures 3b and 3c show zooms of regions respectively in the columnar and equiaxed zone.

Lines 121-123

Samples with a size of 20 x 10 x 10 mm were cut following the blueprint given in Figure3d. Each sample was coated using compression mounting method in a Simplimet 1000 Buehler device.

Lines 180-183

After the normalizing treatment, a ferritic-pearlitic microstructure was expected. Optical micrographs were acquired at different locations in the ingot, either in the center or in the skin part of the cross section. Figure 5 displays two typical examples of the ferritic-pearlitic microstructure observed respectively at the ingot skin an at the ingot center.

Line 383

Figure 13: Comparison of three most common damage variables for all considered strain levels ΔεP/2: 0.01%(a), 0.2% (b), 0.4%(c).